# Optimal Neural Network Approximation for High-Dimensional Continuous Function

## Abstract

The original version of the Kolmogorov-Arnold representation theorem states that for any continuous function $f : [0,1]^d \to \mathbb{R}$, there exist $(d+1)(2d+1)$ univariate continuous functions such that $f$ can be expressed as product and linear combinations of them. So one can use this representation to find the optimum size of a neural network to approximate $f$. Now the important question is to check how does the size of the neural network depends on $d$. It is proved that function space generated by special class of activation function called EUAF (elementary universal activation function), with $36d(2d+1)$ width is dense in $C([a,b]^d)$ with 11 hidden layers. In this paper, we provide classes of $d$-variate functions for which the optimized neural networks will have $\mathcal{O}(d)$ number of neurons with elementary super expressive activation function defined by Yarotsky. We provide a new construction of neural network of $\mathcal{O}(d)$ neuron size to approximate $d$-variate continuous functions of certain classes. We also prove that the size $\mathcal{O}(d)$ is optimal in those cases.

## 1 Introduction

The study of the universal approximation property (UAP) of neural networks has a long-standing history in deep learning. A simple feed forward network is defined by its maximum number of neurons in a hidden layer (called width), number of hidden layers (depth), activation functions used on the neurons on the hidden layers etc. The study can be subdivided in to different parts, where a neural network can have arbitrary width, arbitrary depth, bounded width, bounded depth with different classes of activation functions. The questions that arise in this study are: for which of the property above for a feed forward network, space of continuous functions on a compact set or space of functions on $\mathbb{R}^n$ equipped with various $L^p$ norms (where $1 \le p \le \infty$) can be arbitrarily approximated by the feed forward network with said property.

### 1.1 History

Cybenko (1989) proved that a feed forward network with arbitrary width, bounded depth and sigmoidal activation functions can arbitrarily approximate continuous functions on a compact set in sup norm. When considering the case where the feed forward network has bounded width and arbitrary depth, Lu et al. (2017) proved that networks of width $n+4$ with ReLU activation functions can approximate any Lebesgue integrable function on n-dimensional input space with respect to $L^1$ distance if network depth is allowed to grow. Maiorov and Pinkus (1999) has proved the existence of a network with bounded depth and width that can approximate a continuous function on $[0,1]^d$ within arbitrary error in euclidean norm. All of these results provides existence on such neural networks. There are other results Guliyev and Ismailov (2018a;b) that yields an algorithm on how to choose the weights and activation functions on the neural network to approximate the various spaces of functions. But this problem is difficult due to curse of dimensionality issues. Further research, Hannin and Sellke; Kidger and Lyons (2020); Park et al. (2021), improved the minimum width bound for ReLU networks. Particularly, Park et al. (2021) revealed that the minimum width is $w_{\min} = \max(d_x + 1, d_y)$ for the $L^p(\mathbb{R}^{d_x}, \mathbb{R}^{d_y})$ UAP of ReLU networks and for the $C(K, \mathbb{R}^{d_y})$ UAP of ReLU+STEP networks, where $K$ is a compact domain in $\mathbb{R}^{d_x}$ .

## 1.2 Table

The table below gives summary of known minimum width neural network that has universal approximation properties.

| Functions | Activations | minimum and optimal width | References |
|---|---|---|---|
| $C(K, \mathbb{R})$ | RELU | $w_{\min} = d_x + 1$ | Hannin and Sellke |
| $L^p(\mathbb{R}^{d_x}, \mathbb{R}^{d_y})$ | RELU | $w_{\min} = \max\{d_x + 1, d_y\}$ | Park et al. (2021) |
| $L^p(K, \mathbb{R}^{d_y})$ | Conti. nonpoly | $w_{\min} \leq \max\{d_x + 2, d_y + 1\}$ | Park et al. (2021) |
| $L^p(K, \mathbb{R}^{d_y})$ | Leaky-ReLU | $w_{\min} = \max\{d_x, d_y, 2\}$ | Cai (2023) |
| $L^p(K, \mathbb{R}^{d_y})$ | Leaky-ReLU+ABS | $w_{\min} = \max\{d_x, d_y\}$ | Cai (2023) |
| $C(K, \mathbb{R}_y^d)$ | Arbitrary | $w_{\min} \geq \max\{d_x, d_y\}$ | Cai (2023) |
| $C(K, \mathbb{R}_y^d)$ | ReLU+Floor | $w_{\min} = \max\{d_x, d_y, 2\}$ | Cai (2023) |

## 1.3 Problem and Contribution

1. We provide a theoretical construction of a neural network of width $24d$ and bounded depth that can approximate any $d-$ variate continuous function on $[0, 1]^d$ within arbitrary error in Euclidean norm. So the curse of dimensionality is allayed by our result.

2. We also provide some classes of $d-$ variate continuous functions on $[0, 1]^d$ so that to approximate these classes the width $24d$ of the feed forward network constructed above is minimum.

## 1.4 Related work

One of the ways to obtain the width of the feed forward network that can approximate any $d-$ variate continuous functions on a compact set is to apply the Kolmogorov-Arnold representation theorem. The original version of the Kolmogorov-Arnold Schmidt-Hieber (2021) representation theorem states that for any continuous function $f : [0, 1]^d \to \mathbb{R}$, there exist univariate continuous functions $g_j, h_{i,j}$ such that:

$$f(x_1, x_2, \cdots, x_d) = \sum_{j=0}^{2d} g_j \sum_{i=0}^{d} h_{i,j}.$$

So the expression above implies that we would need a neural network with at most $\mathcal{O}((2d+1)(d+1))$ neurons with some special class of activation functions to approximate a $d$ variate continuous function on $d$-dimensional unit cube. It was shown in Yarotsky (2021) that, for an arbitrary $\epsilon > 0$ and any function $f$ in $C([0, 1]^d)$, there exists a network of size only depending on $d$ constructed with multiple activation functions either ($\sin$ and $\arcsin$ ) or (a non-polynomial analytic function) to approximate $f$ within an error $\epsilon$. The next important question is to check how the neural network size depends on $d$. To approximate the $d$ variate continuous functions as above uniformly, Shen et al. (2022) discovered a special class of activation functions called EUAF (elementary universal activation function). Shen et al. (2022) proved that the function space generated by EUAF networks with width $36d(2d + 1)$, is dense in $C([a, b]^d)$ with 11 hidden layers. Their idea was to prove the above estimation in two steps. In the first step they prove it for one variable and then in the second step they use above Kolmogorov-Arnold representation to come up with precise number of width and depth. In this paper, we improve the number of the neurons a network must have to approximate a $d$ -variate continuous function with bounded depth. Now to get a network with less number of neurons, we can use the theory of space-filling curve. The space filling curve is a continuous surjective function,

$$\ell : [0, 1] \to [0, 1]^d.$$

Then a $d-$ variate continuous function $f$ can be represented as $(f \circ \ell) \circ \ell^{-1}$. Then $f$ can be represented as $g \circ \ell^{-1}$, where $\ell^{-1}$ is a function from $[0, 1]^d$ to $[0, 1]$ and $g = f \circ \ell$. So, $\ell^{-1}$ is a $d-$variate function and $g$ is a one-variate function. So the composition of $g$ and $\ell^{-1}$ would possibly need $\mathcal{O}(d)$ number of neurons with a network with 2 hidden layers. But due to Netto's theorem Kupers we can not possibly have a $\ell^{-1}$ which is an injective continuous function. Therefore, to get around this obstacle, Schmidt-Hieber (2021) discussed a way to replace the inside function as cantor type function. Now in this method one does not require $f$ to be a continuous function, and the inside function which is a cantor type does not need to be a continuous function rather a monotone additive function. The

image of the monotone functions is defined on a Cantor set, $\mathcal{C}$ and the outside function is defined on the Cantor set. Then if $f$ is continuous, then the outside function is also continuous. And to approximate $f$ uniformly, one can approximate the inside function uniformly by continuous function by a deep neural network. The activation functions that can be used for the corresponding deep neural network are Linear activation function and $\mathbb{1}_{x \geq 1/2}$. And one can use combination of RELU function to approximate $\mathbb{1}_{x \geq 1/2}$ as activation function on the deep neural network. Therefore, the above approximation of $d-$ variate continuous function can be approximated efficiently by a deep RELU network. Hannin and Sellke proved the following question regarding parameters needed to approximate continuous function with deep RELU networks:for a fixed $d \geq 1$, what is the minimal width $w$ so that neural nets with ReLU activations, input dimension $d$, hidden layer widths at most $w$, and arbitrary depth can approximate any continuous, real-valued function of $d$ variables arbitrarily well? It turns out that this minimal width is exactly equal to $d + 1$. Now another important direction to allay the curse of dimensionality is to use advanced activation functions. Yarotsky and Zhevnerchuk (2020) showed that (sin,ReLU)-activated neural networks with $W$ parameters can approximate Lipschitz continuous functions with an asymptotic approximation error $\mathcal{O}(e^{-c_d\sqrt{W}})$, where $c_d$ is a constant depending on $d$, the input dimension of the function. Shen et al. (2021a) showed that Lipschitz continuous functions can be approximated by a (Floor,ReLU)-activated neural networks with width $\mathcal{O}(N)$ and depth $\mathcal{O}(L)$ within a error of $\mathcal{O}(\sqrt{d}N^{-\sqrt{L}})$.

## 1.5 ORGANIZATION

Our paper is formulated by the following: in section 2 we derive the uniform approximation of any continuous $d-$variate function on $[0,1]^d$ by a neural network using a composition of functions from the set of super expressive activation functions, using the decomposition of such $d-$variate continuous function by cantor type functions. In section 3 we give examples of continuous functions such that width $d$ is minimum for the above neural network to uniformly approximate these continuous functions. In theorem 2 we can use the super expressive functions defined in Yarotsky (2021), instead of using EUAF defined in Shen et al. (2022) to minimize the total number of parameters in the architecture (not the order). As in (Yarotsky, 2021, Lemma 3) the two variable function $f(x, y) = xy$ is approximated by a network of width 6 and depth 2, on the other hand by using the EUAF the network needed the width 9 and depth 2 (Shen et al., 2022, Lemma 16). Approximation of this multiplier function is an important step to prove the approximation of one variable continuous function using set of activation functions.

## 2 MAIN RESULTS

First we collect the definitions of super-expressive activation functions and EUAF.

Super expressive activation function: Yarotsky (2021) We call a finite family $\mathcal{A}$ of univariate activation functions super-expressive for dimension $d$ if there exists a fixed $d$-input network architecture with each hidden neuron equipped with some fixed activation from the family $\mathcal{A}$, so that any function $f \in C([0,1]^d)$ can be approximated on $[0,1]^d$ with any accuracy in the uniform norm $||\cdot||_\infty$ by such a network, by adjusting the network weights. We call a family A simply super-expressive if it is super-expressive for all $d = 1, 2, ...$ We refer to respective architectures as super-expressive for $\mathcal{A}$. Examples of this type of families are $\mathcal{A}_1 = \{\sigma, \lfloor . \rfloor\}$, $\mathcal{A}_2 = \{\sin, \arcsin\}$ where $\sigma$ is a real analytic and non-polynomial function defined on $(a, b) \in \mathbb{R}$ etc.

EAUF: Shen et al. (2022) A function $\rho : \mathbb{R} \rightarrow \mathbb{R}$ is said to be a universal activation function (UAF) if the function space generated by $\rho$-activated networks with $C_{\rho,d}$ neurons is dense in $C([0,1]^d)$, where $C_{\rho,d}$ is a constant determined by $\rho$ and $d$. An example of this type of family is $\mathcal{A}_3 = \{\sigma_1, \sigma_2\}$, where $\sigma_1$ is a continuous triangular wave function with period 2 i.e.

$$\sigma_1 = \left| x - 2\lfloor \frac{x+1}{2} \rfloor \right|, \quad \forall x \in \mathbb{R}.$$

And $\sigma_2$ is the soft sign activation function i.e.

$$\sigma_2 = \frac{x}{|x|+1}.$$

It is important to note that most of all standard activation functions (except for $\sin$ and $\arcsin$) do not belong to the super-expressive activation family of functions (Yarotsky, 2021, Theorem 5).

We have the following theorem of Schmidt-Hieber (2021).

**Theorem 1.** *Let $F \in C[0,1]^d$, then there exists a continuous function $g : \mathcal{C} \to \mathbb{R}$, where $\mathcal{C}$ is the cantor set and a monotone function $\phi : [0,1] \to \mathcal{C}$ such that:*

$$F(x_1, x_2, \ldots x_d) = g(3 \sum_{j=1}^{d} 3^{-j} \phi(x_j)),$$

*where*

$$\phi(x) = \sum_{i=1}^{\infty} \frac{2a_i^x}{3^{1+d(i-1)}}, \quad x = [0.a_1^x a_2^x \ldots]_2.$$

Now let

$$\phi_k = \sum_{i=1}^{k} \frac{2a_i^x}{3^{1+d(i-1)}},$$

then it can be shown that $\phi_k$ is a continuous function on $[0,1]$. Here $g$ is defined on the cantor set $\mathcal{C}$, which is a closed subspace of the normal space $[0,1]$ so we can extend $g$ to a continuous function $G$ on $[0,1]$ using Tietze extension such that $G|_{\mathcal{C}} = g$. Therefore, using Yarotsky's result we can approximate $g$ with a function from the set of super expressive activation functions, say $N_1$ with an arbitrary approximation error, for this we will need a neural network with width 24. Similarly, let $\phi_k$ be approximated by a function $N_2$ from the set of super expressive activation functions with an arbitrary approximation error. Therefore, we can make the following claim:

**Theorem 2.** *The function $F \in C[0,1]^d$ can be approximated with an arbitrary approximation error by the following composition:*

$$N_1(3 \sum_{j=1}^{d} 3^{-j} N_2(x_j))$$

*Proof.* It will be enough for us to prove that for any $\epsilon > 0$ there exists functions $N_1, N_2$ belonging to the class of super expressive activation functions such that:

$$\left| F(x_1, x_2, \ldots x_d) - N_1(3 \sum_{j=1}^{d} 3^{-j} N_2(x_j)) \right| < \epsilon.$$

Using the triangle inequality we have the following:

$$\left| F(x_1, x_2, \ldots x_d) - N_1(3 \sum_{j=1}^{d} 3^{-j} N_2(x_j)) \right|$$

$$\leq \left| g(3 \sum_{j=1}^{d} 3^{-j} \phi(x_j)) - g(3 \sum_{j=1}^{d} 3^{-j} \phi_k(x_j)) + g(3 \sum_{j=1}^{d} 3^{-j} \phi_k(x_j)) - N_1(3 \sum_{j=1}^{d} 3^{-j} N_2(x_j)) \right|$$

$$\leq \left| g(3 \sum_{j=1}^{d} 3^{-j} \phi(x_j)) - g(3 \sum_{j=1}^{d} 3^{-j} \phi_k(x_j)) \right| + \left| g(3 \sum_{j=1}^{d} 3^{-j} \phi_k(x_j)) - N_1(3 \sum_{j=1}^{d} 3^{-j} N_2(x_j)) \right|.$$

The last step is due to triangle inequality. Now we would like to invoke the uniform continuity of $g$, as $g$ is continuous on compact set $\mathcal{C}$. Therefore, for every $\epsilon > 0$, there exists a $\delta > 0$ such that for all $z_1, z_2 \in \mathcal{C}$, satisfying $|z_1 - z_2| < \delta$ implies $|f(z_1) - f(z_2)| < \epsilon$. Now we approximate the following:

$$3 \sum_{j=1}^{d} 3^{-j} |\phi(x_j) - \phi_k(x_j)| = 3 \sum_{j=1}^{d} 3^{-j} \left( \sum_{i=k+1}^{\infty} \frac{2a_i^{x_j}}{3^{1+d(i-1)}} \right).$$

Then the right-hand side of the above expression is bounded by (as $x_j \in [0, 1]$, for all $j$)

$$3 \sum_{j=1}^{d} 3^{-j} \left( \sum_{i=k+1}^{\infty} \frac{2}{3^{1+d(i-1)}} \right).$$

Now, using the geometric sum we get that the expression above is bounded above by

$$2 \sum_{j=1}^{d} 3^{-j} \frac{1}{3^{kd}} \frac{3}{2}.$$

We again apply the infinite geometric series to get the following expression as an upper bound

$$\frac{1}{2} \frac{1}{3^{dk-1}}.$$

So for a given $\epsilon > 0$ we can always find a $k \in \mathbb{N}$ such that $\frac{1}{2} \frac{1}{3^{dk-1}} < \delta$, so that

$$\left| g(3 \sum_{j=1}^{d} 3^{-j} \phi(x_j)) - g(3 \sum_{j=1}^{d} 3^{-j} \phi_k(x_j)) \right| < \frac{\epsilon}{3}.$$

Now let us investigate the second summand in the above expression. Again using the triangle inequality we get:

$$\left| g(3 \sum_{j=1}^{d} 3^{-j} \phi_k(x_j)) - N_1(3 \sum_{j=1}^{d} 3^{-j} N_2(x_j)) \right|$$

$$\leq \left| g(3 \sum_{j=1}^{d} 3^{-j} \phi_k(x_j)) - g(3 \sum_{j=1}^{d} 3^{-j} N_2(x_j)) \right| + \left| g(3 \sum_{j=1}^{d} 3^{-j} N_2(x_j)) - N_1(3 \sum_{j=1}^{d} 3^{-j} N_2(x_j)) \right|$$

Now for the first summand in the above inequality, we can use the result of Shen et al. (2022) to choose $N_2$ (with the same $\delta > 0$ we used in the previous inequality), such that:

$$|\phi_k(x_j) - N_2(x_j)| < \frac{\delta}{3}.$$

Therefor summing over all $j$ of the above expression, we get:

$$3 \sum_{j=1}^{d} 3^{-j} |\phi_k(x_j) - N_2(x_j)| < \sum_{j=1}^{d} 3^{-j} \delta < \delta.$$

Hence the first summand in the above triangle inequality is less than $\frac{\epsilon}{3}$, using the uniform continuity of $g$.

The second summand of the above triangle inequality is less than $\frac{\epsilon}{3}$, using the result of Shen et al. (2022). Therefore, we arrive at the following approximation:

$$\left| F(x_1, x_2, \ldots x_d) - N_1(3 \sum_{j=1}^{d} 3^{-j} N_2(x_j)) \right| < \epsilon.$$

$\square$

## 2.2 Main theorem

Therefore, using the methods of Shen et al. (2022) on section 5 we have the following theorem:

**Theorem 3.** *Let $F \in C[0, 1]^d$, then $F$ can be approximated within arbitrary error by a function $\Psi$ which is generated by $24d$ width neural network with a combination of super expressive activation functions and EUAF.*

*Proof.* To prove the above theorem we follow the methods of Shen et al. (2022). The key ideas are as follows:

- First prove the theorem for the one-dimensional case. Suppose that an activation function $\zeta$ has a point $x_0$ where the second derivative $\frac{d^2}{dx^2}(\zeta)(x0)$ exists and is nonzero. e.g. we can take $\zeta = \sin(x)$ and $x_0 = \pi k$ for $k$ some integer. This type of activation function is an example of an elementary super expressive activation function defined by Yarotsky (2021). To prove this one-dimensional case we invoke the following results:

  - (Yarotsky, 2021, Lemma 3) For any $M > 0$, there exists a function $\phi$ generated by a network with activation function $\zeta$ and width 6, depth 2 such that

    $$\phi(x,y) = xy, \quad \forall x,y \in [-M, M].$$

  - (Shen et al., 2022, Theorem 14) Let $f \in C([0,1])$ be a continuous function. Given any $\epsilon > 0$, if $K$ is a positive integer satisfying

    $$|f(x_1 - f(x_2))| < \epsilon/2 \quad \forall x_1, x_2 \in [0,1] \quad \text{with} \quad |x_1 - x_2| < 1/K,$$

    then there exists a function $\phi$ generated by an EUAF network with width 2 and depth 3 such that $\|\phi\|_{L^\infty([0,1])} \le \|f\|_{L^\infty([0,1])} + 1$ and

    $$|\phi(x) - f(x)| < \epsilon, \quad \forall x \in \bigcup_{k=0}^{2K}[\frac{2k}{2K}, \frac{2k+1}{2K}].$$

  Now suppose $K \ge 10$. Let $f$ be a uniform continuous function defined on $[-1,1]$ Then we can define $f_i = f(x - i/4K)$ for all $i = 1, 2, 3, 4$ and for all $x \in [0,1]$. Then we apply the second result above to the functions $f_i$. Therefore, for each $i$, a function $\phi_i$ is generated by an EUAF network with width 2 and depth 3. We now construct another function $\psi$ as defined by (Shen et al., 2022, pg. 40) such that our desired $\Psi$ can be written as :

  $$\Psi(x) = \sum_{i=1}^{4} \phi_i(x + i/4K)\psi(2Kx + i/2) \quad \forall x \in [0, 9/10].$$

  Therefore as the inside function in the sum above is a product function, we invoke the first result on each $i$. Hence to construct $\Psi$ we need a $6 \times 4 = 24$ width neural network with a combination of super-expressive activation functions and EUAf. Now as $[0, 9/10]$ can be mapped bijectively to $[0,1]$ we have the desired proof of the theorem.

- Now we invoke Theorem 1 to extend the one-dimensional case to $d$-dimensional case.

$\square$

## 3  COLLECTION OF FUNCTIONS FOR WHICH WIDTH-$d$ IS OPTIMUM

Next, we show that the number of neurons $d$ is optimum. So we will try to construct functions for which we need at least $d$ as suggested by the above analysis.

**Theorem 4.** *Let $F : [0,1]^d \to \mathbb{R}$ be a function which is $d-$linear combination of $\sin$ and, $\cos$ i.e.*

$$F(x_1, x_2, \cdots x_d) = \sum_{k=1}^{d} C_k F_k(x_k),$$

*where $F_k(x_k) = \sin x_k$ or $\cos x_k$. Then we need a neural network of width at least $d$ to approximate this function arbitrarily within $\epsilon > 0$.*

*Proof.* We will first prove it for $d = 2$, then we will prove it for the generel case.

- For d=2, Let $F(x,y) = \cos x + \sin y$. Suppose for any $\epsilon > 0$, there exists $w_1, w_2 \in \mathbb{R}$ and $g$ a continuous (probably composition of functions from EUAF) such that $|F(x,y) - g(w_1x + w_2y)| < \epsilon$. Let $x = y = 0$, then $|1 - g(0)| < \epsilon$. Now assume $y = -\frac{w_1x}{w_2}$ when $w_2 \ne 0$, then $g(0) = \cos x - \sin\frac{w_1x}{w_2}$, which may not be close to 1, not for all $x \in \mathbb{R}$. And when $w_2 = 0$, we can choose $y$ such that, $\sin(y) > 2\epsilon$ so that $|1 + \sin(y) - g(0)| > \epsilon$. If $w_1 = w_2 = w$, then we get that $|1 - g(\pi w/2)| < \epsilon$ and $|g(\pi w/2)| < \epsilon$. Which is contradictory as it will mean $1 - \epsilon < g(\pi w/2) < \epsilon$, for all $\epsilon > 0$.

- For $d = 3$. Let $F(x, y, z) = \sin x + \sin y + \sin z$. Claim is, we can not approximate $F(x, y, z)$ with a width 2 neural network with $\sigma_1, \sigma_2$ being the activation function. Let the function represented by the neural network be

$$a'\sigma_2(\alpha) + b'\sigma_2(\beta),$$

where,

$$\alpha = a\sigma_1(w_1 x + w_2 y + w_3 z) + b\sigma_1(w_1' x + w_2' y + w_3' z)$$

and

$$\beta = a_1\sigma_1(w_1 x + w_2 y + w_3 z) + b_1\sigma_1(w_1' x + w_2' y + w_3' z).$$

Now without loss of generality we can assume that the function above can be written as $g(h(w_1 x + w_2 y + w_3 z), h(w_1' x + w_2' y + w_3' z))$ for some suitable $g, h$, where $h = \sigma_1$. Moreover, we can assume that $h(0) = 0$ Suppose on the contrary we assume that for any $\epsilon > 0$ we have :

$$\left| F(x, y, z) - g(h(w_1 x + w_2 y + w_3 z), h(w_1' x + w_2' y + w_3' z)) \right| < \epsilon,$$

for some $w_1, w_2, w_3, w_1', w_2', w_3' \in \mathbb{R}$. We know that $F(0, 0, 0) = 0$. Therefore, at origin we have that

$$|g(0, 0)| < \epsilon$$

Now the following system :

$$w_1 x + w_2 y + w_3 z = 0, \quad w_1' x + w_2' y + w_3' z = 0,$$

has infinitely many solutions. We can think of $z$ as free variable. Therefore, the solution vector would look like:

$$\left[ \frac{(w_3 w_1' - w_3' w_1)z}{(w_1' w_2 - w_1 w_2')}, \frac{(w_3 w_2' - w_3' w_2)z}{(w_2' w_1 - w_2 w_1')}, z \right]$$

Now when $(w_1' w_2 - w_1 w_2') = 0$, then we have that $\frac{w_1}{w_1'} = \frac{w_2}{w_2'}$. In that case, we will have two free variables. For some fixed, $w_1, w_2, w_3, w_1', w_2', w_3'$ we can choose the value of the free variable such that $|\sin(x) + \sin(y) + \sin(z) - g(0, 0)| > \epsilon$. Therefore we arrive at the contradiction.

- we can extend the above example to $d$ variable function $\sum_{i=1}^{d} \sin(x_i)$. Suppose, on the contrary, we can approximate this function arbitrarily with error $\epsilon > 0$ by

$$f\left( g(\sum_{j=1}^{d-1} w_{1j} x_j), g(\sum_{j=1}^{d-1} w_{2j} x_j), \cdots, g(\sum_{j=1}^{d-1} w_{d-1j} x_j) \right),$$

for some fixed $w_{ij}$, depending on $\epsilon$. We can assume without loss of generality that $g(0) = 0$. Therefore, at the origin, we have $f(0, 0, \cdots, 0) < \epsilon$ Again the solution space of the system:

$$\begin{bmatrix} w_{11} & w_{12} & \cdots w_{1d} \\ \cdots & & \\ w_{d-11} & w_{d-12} & \cdots w_{d-1d} \end{bmatrix} \begin{bmatrix} x_1 \\ x_2 \\ \cdots \\ x_d \end{bmatrix} = 0,$$

will have at least one dimension as a vector space. So we can choose the values of the free variables just like the previous examples such that :

$$\left| \sum_{i=1}^{d} \sin(x_i) - f(0, 0, \cdots, 0) \right| > \epsilon.$$

$\square$

Therefore the functions for which we will need at least $d$ neurons could have the following form $\sum_{i=1}^{d} F_i(x_i)$, where $F_i$'s are continuous functions in one variable and none of them are linear.

**Theorem 5.** *Let $F$ be a $d-$variate continuous real valued function on $[0,1]^d$, such that it has the following form:*

$$F(x_1, x_2, \cdots, x_d) = f(x_1, x_2, \cdots, x_d)(a_1 x_1 + a_2 x_2 + \cdots a_d x_d),$$

*where $f$ is a non-linear, real valued continuous function on $[0,1]^d$ such that it is zero whenever at least one of $x_i = 0$ and $a_i$'s are real numbers. Then we need a neural network of width at least $d$ to approximate this function arbitrarily within $\epsilon > 0$.*

*Proof.* Let us assume by contradiction that the above function $F$ is approximated by a width $d-1$ neural network arbitrarily within error $\epsilon > 0$. Without loss of generality, let the following function represent the neural network for suitable $g$ and $h$:

$$g\left( h(\sum_{j=1}^{d-1} w_{1j} x_j), h(\sum_{j=1}^{d-1} w_{2j} x_j), \cdots, h(\sum_{j=1}^{d-1} w_{d-1j} x_j) \right),$$

for some fixed $w_{ij}$ depending on $\epsilon$. Moreover, we can assume that $g(0) = 0$. Therefore, at the origin we will have $|g(0,0,\cdots,0)| < \epsilon$. Now let $f(x_1, x_2, \cdots, x_d) = 0$, whenever $x_k = 0$, for some fixed $1 \le k \le d$. The solution space of the homogeneous system:

$$\begin{bmatrix} w_{11} & w_{12} & \cdots w_{1d} \\ \cdots \\ w_{d-11} & w_{d-12} & \cdots w_{d-1d} \end{bmatrix} \begin{bmatrix} x_1 \\ x_2 \\ \cdots \\ x_d \end{bmatrix} = 0,$$

will have at least one dimension as a vector space, corresponding to the free variables. If $x_k$ is one of the free variables, then we can always choose the values of the free variables along with $x_k \ne 0$ such that

$$|F(x_1, x_2, \cdots, x_d) - g(0,0,\cdots 0)| > \epsilon.$$

If $x_k$ is not one of the free variables, then $x_k = 0$ can be written as a linear combination of the free variables. If $x_k = 0$, then $F(x_1, x_2, \cdots, x_{k-1}, 0, x_{k+1}, \cdots x_d) = 0$. Now we can always choose the values of the free variables such that :

$$|g\left( h(\sum_{j=1}^{d-1} w_{1j} x_j), h(\sum_{j=1}^{d-1} w_{2j} x_j), \cdots, h(\sum_{j=1}^{d-1} w_{d-1j} x_j) \right)| > \epsilon,$$

hence we arrive at the contradiction. $\qquad\square$

**Corollary 5.1.** *The homogeneous polynomials of degree $k$ of the following form:*

$$\prod_{i=1}^{d} x_i^{k_i}(a_1 x_1 + a_2 x_2 + \cdots a_d x_d),$$

*where $\sum\limits_{i=1}^{d} k_i + 1 = k$ can not be approximated by arbitrarily within $\epsilon > 0$ by a neural network of width $< d$.*

**Theorem 6.** *The homogeneous polynomials of even degree of the following form:*

$$F(x_1, x_2, \cdots, x_d) = \sum_{i=1}^{d} x_i^{2m},$$

*can not be approximated by arbitrarily within $\epsilon > 0$ by a neural network of width $< d$.*

*Proof.* We can follow the same method as above. Therefore, we can choose at least one free variable $x_j$ such that :

$$\sum_{i=1}^{d} a_i^{2m} x_j^{2m} > \epsilon.$$

$\qquad\square$

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
