# OpenReview forum: "Optimal Neural Network Approximation for High-Dimensional Continuous Functions"
_ICLR.cc/2024/Conference — ICLR 2024 Conference Withdrawn Submission_

### Official Review · Reviewer_oDtw · 2023-10-18

**Soundness:** 2 fair
**Presentation:** 1 poor
**Contribution:** 2 fair
**Rating:** 3
**Confidence:** 4

**Summary:**

This paper proposes a universal approximation theorem that uses a $24d$-wide neural network to uniformly approximate a continuous function on a $d$-dimensional compact set arbitrarily well. It is shown that a single-layer neural network needs at least a width of $d$ to be a universal approximator on $[0,1]^d$.

**Strengths:**

* The paper makes the width dependency explicit instead of asymptotic, which is missing from many UAT papers.
* The paper contains a sharpness test. While I believe the test is weak (see the `weaknesses` section), it is a good practice for a theoretical paper.

**Weaknesses:**

* The construction of the universal approximator only has a theoretical meaning. While this is the case for many UAT results, the particular construction in this paper is based on extending a continuous function on a Cantor set to the interval $[0,1]$. This can never be done stably in practice, which impairs the significance of the paper to the ICLR community.
* The work is very incremental. It is a straightforward assembly of Yarotsky's result and Shen's result. It is not made clear in the paper what is the originality of this particular work. While the paper contains in-text proofs, some of the proofs are not self-contained and are hard to interpret without referring to Shen's result.
* The sharpness part of the paper (`section 3`) is questionable. In particular, if I understood correctly, then it is shown that for a single-layer NN, we need at least a width of $d$ to do universal approximation. However, as far as I understood, the UAT result (`Theorem 3`) is for multi-layer NNs. Hence, sharpness is actually not shown. In particular, in a single-layer case, the statement made in `section 3` is almost obvious, because the possible level sets of such a NN form a subspace of dimension $< d$, which cannot be used to do universal approximation.
* The activation functions used in the UAT are not standard in the deep learning society.
* The paper has not been carefully proofread by the author(s). The statements of the theorems are not always clear, and there are a lot of typos. To name a few:
    * The definition of $f$ in section 1.4 does not make sense without variables $x_1, \ldots, x_d$ on the right-hand side.
    * In the second paragraph of  `section 2`, the uniform norm is not properly denoted and the family $\mathcal{A}$ is not in math mode.
    * In the third paragraph of `section 3`, `most of all standard activation functions` is not a standard expression.
    * In the statement of `Theorem 2`, it is unclear what $N_1$ and $N_2$ are without referring to the paragraph before. Then, in the proof, it is said that `It will be enough for us to prove that for any $\epsilon > 0$ there exist functions $N_1, N_2$`. The functions $N_1$ and $N_2$ are prescribed and used in the statement of `Theorem 2`. Why should you find those functions in your proof?
    * In `Theorem 3`, it is unclear what is the quantifier on `with a combination of super expressive activation functions and EUAF`. Do you prefix a set of possible activation functions or it depends on the function you will approximate? Can they be used arbitrarily in the NN you construct, or do you have to use a certain activation function for a specific neuron?
    * In `Theorem 4`, it is never mentioned what the activation functions are.
    * `Therefore` is misspelled on page 5. The condition $d = 2$ is not in math mode in the proof of `Theorem 4`. In the same paragraph, `$g$ a continuous` needs to be corrected. The word ''cannot'' should not be spelled as ''can not.''

**Questions:**

* What is the depth of your NN in each of your theorems?
* What is the particular NN architecture that you have in mind? I assume it is fully-connected feed-forward, but you should have a mathematical formula for your NN in a theoretical paper. In addition, do you have bias terms in the NN and does the last layer have an activation function?
* In the second paragraph of `section 2`, you wrote `each hidden neuron equipped with some fixed activation from the family \mathcal{A}`. Do you mean the activation function is fixed throughout the NN? What about the activation functions in your UAT theorem?
* Not a question but a remark. It is not safe to claim that the UAT result in the paper ''alley the curse of dimensionality,'' because the curse of dimensionality is more about the difficulty in sampling the function than approximating it in theory.

---

### Official Review · Reviewer_ksYY · 2023-10-30

**Soundness:** 3 good
**Presentation:** 3 good
**Contribution:** 2 fair
**Rating:** 5
**Confidence:** 3

**Summary:**

In this paper, the authors study the universal approximation problem of deep neural networks. The main contribution of this paper is to derive that when the input domain is $[0,1]^{d}$, it will require $O(d)$ number of neurons to approximate any continuous function with arbitrary precision for DNNs with a combination of super-expressive activation functions and EUAF (elementary universal activation function). The authors construct neural networks of $O(d)$ neurons to achieve this and show that the bound $O(d)$ is optimal.

**Strengths:**

Originality: The related works are adequately cited. The authors construct neural networks of $O(d)$ neurons to approximate any continuous function with input domain $[0,1]^{d}$ and show that the bound $O(d)$ is optimal. This is an interesting result, which will certainly help us have a better understanding of the universal approximation property of deep neural networks from a theoretical way. I have checked the technique parts and found that the proofs sound solid.

Quality: This paper is technically sound.

Clarity: This paper is clearly written. I find it is easy to follow.

Significance: I think the results in this paper are not very significant, as explained below.

**Weaknesses:**

Although the bound $O(d)$ is optimal and the construction on achieving $O(d)$ neurons is given, I found that the setting of activation functions is artificial in some sense. For example, in the main Theorem 3, the authors require a combination of super-expressive activation
functions and EUAF, which makes the DNN not very practical. It would be interesting to derive the results for one fixed activation
function and for more architectures used in practice.

**Questions:**

It would be interesting to derive the results for one fixed activation function and for more architectures used in practice.

---

### Official Review · Reviewer_j6pw · 2023-10-31

**Soundness:** 1 poor
**Presentation:** 1 poor
**Contribution:** 2 fair
**Rating:** 1
**Confidence:** 3

**Summary:**

Existing studies have shown that if we use the elementary universal activation function (EUAF) as the activation function, there exists a constant depth and width $O(d^2)$ such that for any $\epsilon > 0$, there exists an NN with the specified depth and width that can approximate the true function with an error $\epsilon$. This paper showed that the width can be improved to $O(d)$ by using a super expressive activation function as the activation function. Furthermore, this paper argues that width $O(d)$ is optimal for a particular class of true functions.

**Strengths:**

- Improvement of the width required to approximate the true function from  $O(d^2)$ to $O(d)$ is significant.

**Weaknesses:**

* There is significant room for improvement in the organization and writing of the paper.
* Only a part of the claim in Theorem 4 is proved. Specifically, for general $d$, the proof is only given to the case of the form $\sum_i \sin(x_i)$.
* I have a question about the proof of Theorem 4, as I have a counterexample for it. Let $d=2$, $w_{11} = w_{12} = w$, and $F_1 = F_2 = \sin$. Then, the solution of the system $[w_{11} w_{12}][x_1 x_2]^\top = 0$ is $(x, -x)$. However, when we have $\sin(x) - \sin(x) = 0$.

**Questions:**

* P2, Section 1.4: $\ell^{-1}$ is introduced for the space filling function $\ell$. However, since $\ell$ is only guaranteed to be surjective, $\ell^{-1}$ may not exist.
* In the paragraph between Theorems 1 and 2, Yarotsky's result is cited to show the existence of $N_1$. On the other hand, Shen's result is used in the proof of Theorem 2. I want to clarify which is correct.
* For the proof of Theorem 2, only a proof sketch is given. The proof should be mathematically rigorous enough.

【Minor Comments】
* Abstract: *It is proved that function space [...] with 11 hidden layers.*: When I read this part for the first time, I thought this was the paper's contribution. However, this results from an existing study and should be clearly stated as such.
* Section 1.2: I suggest that the section title is aligned with the content (e.g., Known UAP Results)
* Section 1.4: It is unclear what the expressions *inside/outside function* refers to. I suggest reconsidering the expression.
* Section 2: $\|\_\infty$ -> $\|\cdot\|\_\infty$
* Section 2: A -> $\mathcal{A}$
* Section 2.1: Therefore -> Therefore
* Section 2.2: $|f(x_1 - f(x_2))| < \epsilon/2$ -> $|f(x_1) - f(x_2)| < \epsilon/2$
* Section 2.2: EUAf -> EUAF
* Section 4: Section titles are duplicated.
* Some references do not mention the publication year (e.g., Hannin and Sellke in Section 1.2, Kupers in Section 1.4.)

**Details Of Ethics Concerns:**

N.A.

---

### Official Review · Reviewer_jmHW · 2023-11-02

**Soundness:** 2 fair
**Presentation:** 1 poor
**Contribution:** 1 poor
**Rating:** 1
**Confidence:** 2

**Summary:**

In this paper, the authors consider computing the optimal width for neural networks of bounded depth to approximate functions in $[0,1]^d$. Considering a special class of activation functions, a set of super expressive activation functions defined by Yarotsky, they construct a neural network of bounded depth with $24d$ width that can approximate any continuous functions to arbitrary error in sup norm. They further provide examples of multivariate functions where width $d$ is minimal for the neural networks to approximate these functions.

**Strengths:**

- The construction is simple to follow and the statements of the theorems are somewhat easy to understand.

**Weaknesses:**

- It seems that the theorems are obtained by combining previously obtained results by Shen et al., and Yarotsky, and there are not many innovations. Per se, this is not necessarily bad, but combined with the other weaknesses, it makes this paper much less appealing.
- Is a minimum $d$ width not trivial? If we only have access to $k<d$ ridge functions $\sigma (\< w_k , x \>)$, then the neural network only depend son a $k$-dimensional projection of the data and none of section 3 seems to be necessary.
- The problem is very niche, and seems to be a purely mathematical curiosity rather than anything relevant for practice. Non-standard activations and small width neural networks are hard to train. Hence, I wonder how much the ICLR community will enjoy that paper.
- The paper seems to have been written in a rush. There are many typos, either in the text or in the equations. The introduction, discussions and related work are poorly written and very hard to follow. A lot of statements are imprecise, which makes understanding the motivations, difficulties and achievements hard to parse for someone that is not already well aware with this literature. Overall the paper is extremely difficult to read because of all these points. I encourage the authors to dedicate a lot of time improving the writing and clarity of the text, possibly with the help of a writing assistant software.

**Questions:**

See weaknesses:
- Improve the text and clarity.
- Is Section 3 not trivial?
- Is there a substantial technical contribution?

---

### Meta-Review · Area_Chair_CRxu · 2023-12-04

**Metareview:**

This paper is a clear reject. The paper had been reviewed before and reviewer comments have not been addressed. In particular, one of the former reviewers provided counter-examples to claims in the paper and these have not been commented on/have not been addressed. I suggest that the authors take reviewer comments into account for future submissions and do not resubmit the paper with disproven claims in it.

**Justification For Why Not Higher Score:**

n/a

**Justification For Why Not Lower Score:**

n/a

---

### Decision · Program_Chairs · 2024-01-16

Reject